# Interaction of wheat cultivar and enzyme on broiler growth, nutrient utilization, and gut microflora

Seyedkamyar Seyedoshohadaei[1], Mehran Torki[1]*, Akbar Yaghoubfar[2], Alireza Abdolmohammadi[1]

1 Animal Science Department, College of Agriculture and Natural Resources, Razi University, Kermanshah, Iran, 2 Hyderabad Animal Science Research Institute, Karaj, Iran

* torki@razi.ac.ir

**Data Availability Statement:** All relevant data are within the manuscript and its Supporting Information files.

## Abstract

This study examined the impacts of four wheat cultivars and enzyme supplementation on growth performance, nutrient digestibility, and ileal microbiota composition in broiler chickens. Six hundred forty-eight male broilers (1-day-old, Ross 308) were studied in a completely randomized design factorial 4 × 2 along with control (9 treatments) with 6 replications (12 birds per pen). The Diets consisted of the four varieties of wheat (Sardari, Azar2, Sirvan, and Pishgam) with and without enzyme supplementation, alongside a corn-based control diet. All diets were iso-caloric and iso-nitrogenous. Daily weight gain (DWG) and feed conversion ratio (FCR) were not significantly affected by the dietary treatments. Broilers fed the corn-based diet displayed higher feed intake (FI) than those fed diets containing different wheat cultivars. Enzyme supplementation in wheat-based diets did not impact broiler growth performance. There was an interaction between enzyme and wheat type for protein, fat, calcium, and phosphorus digestibility. Ileal microbiota analysis revealed no significant changes in *Lactobacillus* and *Escherichia coli* populations across treatments. Conversely, *Enterococcus* and *Bifidobacteria* populations exhibited significant differences, with the Sirvan cultivar diet promoting the highest bacterial counts. It was concluded that different wheat cultivars could affect growth performance, nutrient digestibility, and ileum microbiota, and the beneficial effect of supplemental enzymes was only evident in certain variables and depended on the specific wheat variety.

## Introduction

Rising corn prices are squeezing profit margins in the poultry industry, where feed costs account for a significant portion (60–70%) of production expenses. This economic pressure necessitates exploring alternative feed sources that are both nutritious and economical. Wheat emerges as a promising candidate due to its abundance and affordability in certain regions [1]. However, its chemical composition, particularly the presence of non-starch polysaccharides (NSPs), presents challenges. Unlike corn, wheat contains high levels of NSPs, especially

**Funding:** The author(s) received no specific funding for this work.

**Competing interests:** The authors have declared that no competing interests exist.

arabinoxylans, which limit nutrient utilization in poultry [2, 3]. These complex carbohydrates create a viscous gut environment, hindering digestive enzyme activity and nutrient absorption [4, 5]. Furthermore, undigested NSPs can be excessively fermented by gut microbes, potentially disrupting the intestinal microbial balance and hindering further nutrient uptake [6]. Broilers lack sufficient xylanase enzymes required to break down these NSPs, leading to performance issues, particularly in young birds [7]. Fortunately, incorporating exogenous enzymes into wheat-based diets has shown promise in addressing these challenges [8]. Studies suggest that enzymes like xylanase can improve nutrient digestibility, reduce gut viscosity, and modify gut microbiota composition, ultimately benefiting broiler performance [9–11]. However, the effectiveness of enzyme supplementation can vary depending on the specific wheat cultivar used in the diet [12]. Additionally, factors like bird age, breed, and the type of enzyme employed can influence the observed response [1, 13]. Given the variability in wheat composition and the potential benefits of enzyme supplementation, this study investigates the combined effects of different wheat cultivars and enzyme inclusion on broiler performance, nutrient digestibility, and gut microflora.

## Materials and methods

### Ethics statement

This experiment received approval from Razi University's Animal Ethics Committee in Kermanshah, Iran, ensuring all procedures followed strict animal welfare guidelines (Code Number: Anim Sci, 1395, # 23). These protocols adhere to European Union standards for animal protection and feed legislation.

### Broiler chickens, management and diets

The experiment involved 648 one-day-old chicks of the Ross 308 broiler breed. Chicks were first sexed in the incubator using the feathering method and then randomly distributed across 54 pens (9 treatments × 6 replicates) with 12 chicks per pen. Each pen measured 1.20 meters by 1.50 meters, providing adequate space for the birds. Throughout the study, the broilers had ad libitum access to water and feed for optimal health and growth. Vaccination schedules, brooding temperatures and humidity levels, and nutritional requirements strictly followed the Ross 308 strain breeding guide. A completely randomized 2 × 4 factorial design with a control treatment was employed to investigate the impact of wheat cultivars (Sardari, Azar2, Sirvan, and Pishgam) and enzyme supplementation (with and without) on broiler performance. Detailed compositions of these diets are presented in Tables 1 and 2. It's important to note that all diets were balanced to have the same levels of energy, protein, and dietary electrolyte balance, allowing for a direct comparison of the effects of wheat cultivar and enzyme supplementation.

### Wheat cultivars and enzyme

Four Iranian wheat cultivars were used in this experiment: Sardari, Azar 2, Pishgam, and Sirvan. These cultivars were obtained from wheat distribution companies operating under the supervision of the Agricultural Research Institutes of Kurdistan and Kermanshah Provinces. A commercial multi-enzyme product named beta-endopower® (Easy Bio, Inc., Korea) was used as the enzyme supplement. This product is a combination of alpha-galactosidase, galactomannanase, xylanase, and beta-glucanase activities, produced through solid-state fermentation of non-GMO *Aspergillus niger* and *Aspergillus oryzae* fungi. The total enzyme concentration in the feed was 0.2 g/kg. For reference, the enzyme activities included at least 1,500 (units/g) of

**Table 1. Ingredient composition of basal diet and experimental diets (starter and grower).**

| Ingredients, % | Starter | | | | | Grower | | | | |
|---|---|---|---|---|---|---|---|---|---|---|
| Corn | 41.87 | - | - | - | - | 45.93 | - | - | - | - |
| Sardari | - | 47.47 | - | - | - | - | 51.56 | - | - | - |
| Azar2 | - | - | 46.83 | - | - | - | - | 51.60 | - | - |
| Pishgam | - | - | - | 46.98 | - | - | - | - | 51.46 | - |
| Sirvan | - | - | - | - | 50.10 | - | - | - | - | 53.91 |
| Soybean meal | 43.15 | 43.57 | 44.40 | 44.13 | 39.69 | 37.82 | 38.21 | 38.74 | 38.88 | 33.98 |
| Wheat bran | 6 | - | - | - | 1.40 | 6.64 | 0.56 | - | - | 2.38 |
| Vegetable oil | 4.5 | 4.62 | 4.40 | 4.53 | 4.32 | 5.50 | 5.50 | 5.50 | 5.50 | 5.50 |
| Calcium carbonate | 0.67 | 0.6 | 0.6 | 0.6 | 0.59 | 0.57 | 0.51 | 0.51 | 0.51 | 0.49 |
| Di-calcium phosphate | 1.7 | 1.67 | 1.68 | 1.68 | 1.74 | 1.51 | 1.49 | 1.49 | 1.49 | 1.55 |
| Oyster shell | 0.5 | 0.5 | 0.5 | 0.5 | 0.5 | 0.50 | 0.50 | 0.50 | 0.50 | 0.50 |
| Salt | 0.5 | 0.5 | 0.5 | 0.5 | 0.5 | 0.50 | 0.50 | 0.50 | 0.50 | 0.50 |
| Vitamin premix[1] | 0.25 | 0.25 | 0.25 | 0.25 | 0.25 | 0.25 | 0.25 | 0.25 | 0.25 | 0.25 |
| Mineral premix[2] | 0.25 | 0.25 | 0.25 | 0.25 | 0.25 | 0.25 | 0.25 | 0.25 | 0.25 | 0.25 |
| L-Lysine | 0.22 | 0.20 | 0.21 | 0.20 | 0.23 | 0.21 | 0.20 | 0.18 | 0.19 | 0.22 |
| DL-Methionine | 0.31 | 0.30 | 0.30 | 0.30 | 0.31 | 0.27 | 0.26 | 0.27 | 0.26 | 0.27 |
| L-Threonine | 0.08 | 0.07 | 0.08 | 0.08 | 0.12 | 0.05 | 0.21 | 0.21 | 0.21 | 0.20 |
| *Calculated analysis (%)* | | | | | | | | | | |
| ME (kcal/kg) | 2900 | 2900 | 2900 | 2900 | 2900 | 3178 | 3178 | 3178 | 3178 | 3178 |
| Crude protein | 24 | 24 | 24 | 24 | 24 | 22.04 | 22.04 | 22.04 | 22.04 | 22.04 |
| Ether Extract | 6.2 | 5.3 | 5.6 | 5.47 | 5.45 | 7.20 | 6.30 | 6.45 | 6.40 | 6.48 |
| Crude Fiber | 4.6 | 4.1 | 4.4 | 4.26 | 4.1 | 4.39 | 3.78 | 4.11 | 4.01 | 3.93 |
| Calcium | 0.96 | 0.96 | 0.96 | 0.96 | 0.96 | 0.78 | 0.78 | 0.78 | 0.78 | 0.78 |
| Available Phosphorus | 0.48 | 0.48 | 0.48 | 0.48 | 0.48 | 0.435 | 0.435 | 0.435 | 0.435 | 0.435 |
| Na | 0.22 | 0.23 | 0.23 | 0.23 | 0.23 | 0.22 | 0.23 | 0.23 | 0.23 | 0.23 |
| Lysine | 1.32 | 1.33 | 1.34 | 1.33 | 1.28 | 1.20 | 1.21 | 1.20 | 1.21 | 1.16 |
| Methionine | 0.61 | 0.59 | 0.60 | 0.60 | 0.60 | 0.56 | 0.54 | 0.54 | 0.54 | 0.54 |
| Threonine | 0.86 | 0.86 | 0.86 | 0.86 | 0.86 | 0.77 | 0.87 | 0.87 | 0.80 | 0.80 |

[1] Vitamin premix provided the following (per kg of diet): thiamin-mononitrate, 2.4 mg; nicotinic acid, 44 mg; riboflavin, 4.4 mg; D- Ca pantothenate, 12 mg; vitamin B12 (cobalamin), 12.0 mg; pyridoxine HCL, 4.7 mg; D-biotin, 0.11 mg; folic acid, 5.5 mg; menadione sodium bisulfate complex, 3.34 mg; choline chloride, 220 mg; cholecalciferol, 27.5 mg; transretinyl acetate, 1892 mg; all-rac α tocopheryl acetate, 11 mg; ethoxyquin, 125 mg.

[2] Mineral premix provided the following (per kg of diet): manganese (MnSO4-H2O), 60 mg; iron (FeSO4-7H2O), 30 mg; zinc (ZnO), 50 mg; copper (CuSO4-5H2O), 5 mg; iodine (ethylene diaminedihydroiodide), 0.15 mg; selenium (NaSe03), 0.3 mg.

xylanase, 1,100 (units/g) of beta-glucanase, 110 (units/g) of galactomannanase, and 35 (units/g) of alpha-galactosidase.

## Chemical analysis

Before the experiment, the chemical composition of wheat cultivars and experimental diets was analyzed (Table 3). Dry matter (DM), crude protein (CP), crude fiber, crude fat, and ash content were determined in all samples (diets, wheat cultivars, and ileal samples) following the procedures outlined by the AOAC [14]. Additionally, sugar and starch content in the wheat cultivars were measured [14]. Furthermore, acid detergent fiber (ADF) and neutral detergent fiber (NDF) content were assessed in the samples using the method described by Van Soest et al. [15]. The Megazyme assay kits (K-TDFR, Ireland) based on AACC (American Association of Cereal Chemists) method 32–07.01 and AOAC methods 991.43 were employed to

**Table 2. Ingredient composition of basal diet and experiment diets (finisher).**

| Ingredients, % | Finisher | | | | |
|---|---|---|---|---|---|
| Corn | 49.64 | - | - | - | - |
| Sardari | - | 55.90 | - | - | - |
| Azar2 | - | - | 55.73 | - | - |
| Pishgam | - | - | - | 55.74 | - |
| Sirvan | - | - | - | - | 58.37 |
| Soybean meal | 30.86 | 31.31 | 32.40 | 32.01 | 26.71 |
| Wheat bran | 11.34 | 4.67 | 3.77 | 4.15 | 6.74 |
| Vegetable oil | 4.37 | 4.37 | 4.37 | 4.37 | 4.37 |
| Calcium carbonate | 0.49 | 0.42 | 0.42 | 0.42 | 0.40 |
| Di-calcium phosphate | 1.30 | 1.28 | 1.28 | 1.28 | 1.34 |
| Oyster shell | 0.50 | 0.50 | 0.50 | 0.50 | 0.50 |
| Salt | 0.50 | 0.50 | 0.50 | 0.50 | 0.50 |
| Vitamin premix[1] | 0.25 | 0.25 | 0.25 | 0.25 | 0.25 |
| Mineral premix[2] | 0.25 | 0.25 | 0.25 | 0.25 | 0.25 |
| L-Lysine | 0.21 | 0.21 | 0.19 | 0.20 | 0.23 |
| DL-Methionine | 0.24 | 0.24 | 0.24 | 0.24 | 0.24 |
| L-Threonine | 0.05 | 0.10 | 0.10 | 0.09 | 0.10 |
| *Calculated analysis*, % | | | | | |
| ME (kcal/kg) | 3000 | 3000 | 3000 | 3000 | 3000 |
| Crude protein | 20 | 19.64 | 19.64 | 19.64 | 19.64 |
| Ether Extract | 6.4 | 6.4 | 6.4 | 6.4 | 6.4 |
| Crude Fiber | 4.5 | 4.5 | 4.5 | 4.5 | 4.5 |
| Calcium | 0.78 | 0.78 | 0.78 | 0.78 | 0.78 |
| Available Phosphorus | 0.39 | 0.39 | 0.39 | 0.39 | 0.39 |
| Na | 0.22 | 0.22 | 0.22 | 0.22 | 0.22 |
| Lysine | 1.05 | 1.05 | 1.05 | 1.05 | 1.05 |
| Methionine | 0.51 | 0.51 | 0.51 | 0.51 | 0.51 |
| Threonine | 0.81 | 0.81 | 0.81 | 0.81 | 0.81 |

[1] Vitamin mix provided the following (per kg of diet): thiamin-mononitrate, 2.4 mg; nicotinic acid, 44 mg; riboflavin, 4.4 mg; D- Ca pantothenate, 12 mg; vitamin B12 (cobalamin), 12.0 mg; pyridoxine HCL, 4.7 mg; D-biotin, 0.11 mg; folic acid, 5.5 mg; menadione sodium bisulfate complex, 3.34 mg; choline chloride, 220 mg; cholecalciferol, 27.5 mg; transretinyl acetate, 1892 mg; all-rac α tocopheryl acetate, 11 mg; ethoxyquin, 125 mg.

[2] Trace mineral mix provided the following (per kg of diet): manganese ($MnSO_4-H_2O$), 60 mg; iron ($FeSO_4-7H_2O$), 30 mg; zinc (ZnO), 50 mg; copper ($CuSO_4-5H_2O$), 5 mg; iodine (ethylene diaminedihydroiodide), 0.15 mg; selenium ($NaSe0_3$), 0.3 mg.

measure soluble and insoluble fiber. NSP content was determined according to the methods established by Kalantar and Yaghobfar [16]. Amylose and amylopectin content were measured using the specific Megazyme kit K-AMYL. Gross energy was measured using a Parr 1261 bomb calorimeter standardized with benzoic acid. Calcium and phosphorus content in ileal samples and wheat cultivars were analyzed by first ashing and digesting the samples according to AOAC [17] methods, followed by measurement with a Jenway Model 6300 spectrophotometer (USA).

## Growth performance

The study evaluated broiler chicken performance through daily weight gain (DWG), feed intake (FI), and feed conversion ratio (FCR) from 0 to 39 days of age.

**Table 3. Chemical composition of different wheat cultivars (% DM).**

| Chemical composition (% DM) | Wheat cultivars | | | |
|---|---|---|---|---|
| Parameters | Sardari | Azar 2 | Pishgam | Sirvan |
| DM | 94.18 | 94.34 | 96.61 | 96.61 |
| CP | 9.72 | 9.31 | 14.07 | 15.27 |
| Crude Fiber | 3.00 | 4.00 | 2.40 | 2.80 |
| NFE | 78.48 | 76.04 | 75.98 | 75.13 |
| NDF | 12.75 | 13.50 | 11.25 | 15.75 |
| ADF | 2.75 | 2.50 | 2.25 | 3.50 |
| ADL | 8.75 | 9.90 | 8.30 | 11.40 |
| Ash | 1.50 | 1.40 | 1.55 | 2.65 |
| Crude Fat | 1.30 | 2.00 | 2.50 | 1.90 |
| Gross Energy (cal/g) | 4125.25 | 4128.11 | 4082.75 | 4003.36 |
| Calcium | 0.42 | 0.30 | 0.18 | 0.28 |
| Available Phosphorus | 0.25 | 0.22 | 0.15 | 0.12 |
| Sugar | 5.31 | 6.06 | 5.17 | 5.58 |
| Starch | 74.95 | 68.73 | 75.88 | 56.75 |
| Insoluble Fiber | 18.367 | 12.483 | 16.622 | 19.712 |
| Soluble Fiber | 4.110 | 4.568 | 6.077 | 8.292 |
| NSP | 11.75 | 12.70 | 10.70 | 15.4 |
| Amylose | 7.29 | 4.80 | 6.61 | 6.69 |
| Amylopectin | 92.71 | 95.20 | 93.39 | 93.31 |

DM: dry matter, CP: Crude protein, NFE: Nitrogen Free Extract, NDF: Neutral Detergent Fiber, ADF: Acid Detergent Fiber, ADL: Acid Detergent Lignin, NSP: non-starch polysaccharides, Cellulose = NDF–ADF, Cellulose = ADF–(ADL + Ash), NSP = Crude Fiber + ADL (Kalantar & Yaghobfar, 2016)

## Nutrient digestibility assay

On the 38th day of the experiment, nutrient digestibility was evaluated. A random selection of 54 broilers (six replicates from each of the nine treatments) were separated from the main flocks and fed diets containing titanium dioxide (3 g/kg) as an indigestible marker for 72 hours. Following this feeding period, all chickens were humanely slaughtered, and their small intestines were collected. Ileal digesta was carefully removed by hand pressure, collected in falcon tubes, and stored at -20°C. The collected samples were then dried at 60°C and ground for further analysis. This analysis determined the digestibility of DM, CP, crude fat, ash, calcium, and phosphorus. Additionally, the concentration of titanium dioxide in both the ileal digesta and the experimental diets was examined according to the method described by Short et al. [18]. This marker allowed researchers to calculate nutrient digestibility coefficients.

## Microbial analysis

On day 39, three birds from each replicate were anaesthetized via inhalation of 60% $CO_2$ and then sacrificed by cervical dislocation. $CO_2$ was delivered from the compressed gas canister and delivered using a gradual fill method with a displacement rate of 60% via the chamber volume per minute using a flowmeter. Then their ileal digesta was collected in sterile bottles and stored frozen at -20°C until further analysis. The microbial populations of *Lactobacillus*, *Escherichia coli* (*E. coli*), *Enterococci*, and *Bifidobacteria* were assessed using the method described by Zhang et al. [19]. Briefly, the ileal digesta samples were serially diluted (10-fold) in a sterile sodium chloride solution to create dilutions of $10^{-1}$, $10^{-2}$, and $10^{-3}$. Three replicates of each dilution were then cultured on specific media to enumerate the targeted bacteria: MRS agar for *Lactobacillus*, MacConkey agar for *E. coli*, MRS agar supplemented with 40%

tomato pulp for *Bifidobacteria*, Nutrient agar for *Enterococci*. Following incubation at 37°C for 24 hours, the number of bacterial colonies formed in the digesta samples were counted. These counts were then converted to log CFU per gram of digesta content.

## Calculations and statistical analysis

The apparent ileal digestibility of DM, CP, crude fat, ash, calcium, and phosphorus was calculated by the following formula:

Apparent ileal digestibility % = [(NT/Ti)$_{diet}$- (NT/Ti)$_{ileal\ digesta}$]/ (NT/Ti)$_{diet}$*100

where (NT/Ti) $_{diet}$ = ratio of component and titanium in the diet, and (NT/Ti) $_{ileal\ digesta}$ = ratio of component and titanium in ileal digesta.

Data were analyzed using SAS software version 9.4 [20] following a completely randomized $4 \times 2$ factorial design. This design evaluated the main effects of the wheat cultivar (4 levels) and enzyme supplementation (2 levels), as well as their interaction, on the measured characteristics. The 9th treatment was the corn-based control diet, which was not part of the factorial analysis but included for comparison. The statistical model for the factorial design can be expressed as: $Y_{ijk} = \mu + A_i + B_j + AB_{ij} + e_{ijk}$, where: $Y_{ijk}$ = measured characteristic, $\mu$ = overall mean, $A_i$ = main effect of the ith wheat cultivar $B_j$ = main effect of the jth enzyme level $AB_{ij}$ = interaction effect between the ith wheat cultivar and jth enzyme level $e_{ijk}$ = residual error.

A comparison of mean values among treatments was conducted using the least significant difference (LSD) test. If a significant interaction ($AB_{ij}$) was detected, the main effects ($A_i$ and $B_j$) were not interpreted independently ($p < 0.05$). For comparison of means across all nine experimental treatments, a completely randomized design model was employed: $Y_{ij} = \mu + T_i + e_i$, where:

$Y_{ij}$ = measured characteristic, $\mu$ = overall mean, $T_i$ = effect of the ith treatment, $e_i$ = residual error.

## Results

### Growth performance

The effects of dietary treatments on growth performance parameters (FI, FCR, and DWG) of broilers from 0 to 39 days of age are presented in Table 4. There were no remarkable differences ($p > 0.05$) observed in DWG and FCR across the nine experimental diets. However, FI displayed significant variation among treatments ($p < 0.05$). Broilers fed the corn-based diet exhibited the highest FI compared to all other dietary treatments, except for those containing the Sardari wheat cultivar with enzyme supplementation ($p < 0.05$).

### Nutrient digestibility

Table 5 explores the impact of different wheat cultivars (with and without enzyme supplementation) on the ileal digestibility of various nutrients in broilers. There was no significant overall effect on DM and ash digestibility due to wheat cultivar or enzyme presence ($p > 0.05$). However, significant interactions ($p < 0.05$) were observed between wheat type and enzyme supplementation for CP, fat, calcium, and phosphorus digestibility. For CP digestibility, the Sardari cultivar without enzyme stood out with the highest value, significantly different from other diets ($p < 0.05$). Interestingly, enzyme addition generally decreased CP digestibility across cultivars, except for Sirvan, where it significantly increased ($p < 0.05$). Similar trends emerged for fat digestibility. The Pishgam cultivar without enzyme showed the highest digestibility, while Sirvan without enzyme and corn diets had lower values ($p < 0.05$). Adding enzymes significantly boosted fat digestibility in Sardari and Sirvan cultivars compared to other treatments ($p < 0.05$). Phosphorus digestibility displayed a similar pattern. Diets containing the

**Table 4. The effect of wheat cultivars with and without enzyme and control diet on performance parameters in broilers.**

| | Daily weight gain, g /chick/ day | Feed intake, g /chick | Feed conversion ratio, g /g |
|---|---|---|---|
| | (Day 0–39) | (Day 0–39) | (Day 0–39) |
| **WC** | | | |
| Sardari | 52.78 ± 3.04 | 75.31 ± 3.25 | 1.43 ± 0.06 |
| Azar 2 | 51.12 ± 2.40 | 73.73 ± 2.65 | 1.44 ± 0.07 |
| Pishgam | 50.13 ± 3.44 | 73.56 ± 3.04 | 1.47 ± 0.07 |
| Sirvan | 49.67 ± 3.20 | 74.09 ± 3.08 | 1.49 ± 0.09 |
| p-value | 0.080 | 0.478 | 0.210 |
| SEM | 0.884 | 0.860 | 0.023 |
| **E** | | | |
| Zero | 50.24 ± 3.41 | 73.49 ± 3.45 | 1.46 ± 0.08 |
| 0.2, g/Kg | 51.61 ± 2.84 | 74.85 ± 2.34 | 1.45 ± 0.07 |
| p-value | 0.129 | 0.121 | 0.556 |
| SEM | 0.625 | 0.608 | 0.016 |
| **WC × E** | | | |
| p-value | 0.745 | 0.468 | 0.698 |
| CV | 6.01 | 4.01 | 5.50 |
| Treatments | | | |
| **WC          E** | | | |
| Sardari— | 52.41 ± 3.84 | 74.31 ± 3.71 [bc] | 1.42 ± 0.06 |
| Sardari + | 53.15 ± 2.31 | 76.32 ± 2.65 [ab] | 1.43 ± 0.07 |
| Azar 2 - | 50.06 ± 2.16 | 72.17 ± 2.85 [c] | 1.44 ± 0.10 |
| Azar 2 + | 52.18 ± 2.31 | 75.29 ± 1.20 [bc] | 1.44 ± 0.05 |
| Pishgam - | 50.08 ± 3.51 | 73.88 ± 3.80 [bc] | 1.47 ± 0.05 |
| Pishgam + | 50.18 ± 3.71 | 73.25 ± 2.36 [bc] | 1.46 ± 0.09 |
| Sirvan - | 48.41 ± 3.50 | 73.62 ± 3.90 [bc] | 1.52 ± 0.07 |
| Sirvan + | 50.93 ± 2.55 | 74.57 ± 2.29 [bc] | 1.46 ± 0.10 |
| Corn | 52.10 ± 1.33 | 78.77 ± 2.88 [a] | 1.51 ± 0.05 |
| p-value | 0.151 | 0.021 | 0.226 |
| SEM | 1.19 | 1.21 | 0.031 |

WC, wheat cultivar; E, enzyme

[a-c] Means in a column, within a group, but with different superscripts are significantly ($p < 0.05$) different.

Values represent means ± standard deviation of the mean.

SEM, standard error of the mean.

unenzymed Sardari cultivar, Sirvan with enzyme, and corn had the highest values. The Azar 2 cultivar without enzyme exhibited the lowest digestibility. Fortunately, adding enzymes to Azar 2, Pishgam, and Sirvan diets significantly improved their phosphorus digestibility ($p < 0.05$). Calcium digestibility results differed slightly. The unenzymed Sardari diet had the highest value, while the Pishgam diet with enzyme and the unenzymed and enzyme-supplemented Sirvan diets displayed the lowest. Notably, enzyme addition reduced calcium digestibility in the Sardari and Pishgam cultivar diets ($p < 0.05$).

## Intestinal microflora

Table 6 details the impact of dietary treatments on ileal microbiota populations in broilers. There was no significant change ($p > 0.05$) observed in *Lactobacillus* or *E. coli* populations across the various treatments. However, wheat cultivars themselves significantly affected

**Table 5. The effect of wheat cultivars with and without enzyme and control diet on digestibility (%) of nutrients in broilers.**

| | Digestibility, % | | | | | |
|---|---|---|---|---|---|---|
| | Dry matter | Ash | Protein | Fat | Phosphorus | Calcium |
| **WC** | | | | | | |
| Sardari | 68.75 ± 3.73 | 64.68 ± 18.88 | 71.45 ± 11.55[a] | 59.66 ± 14.25[b] | 47.62 ± 4.12[a] | 48.70 ± 11.84[a] |
| Azar 2 | 68.56 ± 4.33 | 71.46 ± 18.26 | 57.65 ± 8.79[c] | 69.44 ± 10.69[ab] | 25.12 ± 4.00[d] | 46.70 ± 6.35[a] |
| Pishgam | 64.28 ± 9.45 | 82.23 ± 9.87 | 54.55 ± 7.36[c] | 77.92 ± 9.32[a] | 32.32 ± 7.82[c] | 36.54 ± 19.61[b] |
| Sirvan | 73.83 ± 5.43 | 82.88 ± 6.23 | 65.01 ± 6.09[b] | 47.06 ± 16.07[c] | 39.58 ± 9.48[b] | 27.02 ± 3.39[c] |
| p-value | 0.114 | 0.092 | 0.0001 | 0.0003 | 0.0001 | 0.0001 |
| SEM | 2.56 | 5.51 | 1.60 | 3.91 | 1.34 | 2.63 |
| **E** | | | | | | |
| Zero | 68.09 ± 5.58 | 71.18 ± 19.26 | 66.41 ± 10.41[a] | 59.97 ± 21.39 | 32.05 ± 11.36[a] | 46.96 ± 14.77[a] |
| 0.2, g/Kg | 69.62 ± 8.73 | 79.44 ± 9.72 | 57.91 ± 9.15[b] | 67.07 ± 10.21 | 40.27 ± 8.33[b] | 32.53 ± 9.72[b] |
| p-value | 0.558 | 0.153 | 0.0001 | 0.088 | 0.0001 | 0.0001 |
| SEM | 1.81 | 3.90 | 1.13 | 2.76 | 0.949 | 1.86 |
| **WC × E** | | | | | | |
| p-value | 0.437 | 0.263 | 0.0001 | 0.008 | 0.0005 | 0.002 |
| CV | 9.12 | 17.95 | 6.30 | 15.09 | 9.09 | 16.22 |
| Treatments | | | | | | |
| **WC          E** | | | | | | |
| Sardari— | 69.49 ± 2.38 | 58.02 ± 23.45 | 81.89 ± 1.44[a] | 47.84 ± 4.31[cd] | 49.36 ± 2.15[a] | 59.05 ± 4.48 [a] |
| Sardari + | 68.00 ± 5.24 | 71.33 ± 14.44 | 61.00 ± 2.06[c] | 71.48 ± 8.40[ab] | 45.88 ± 5.36[ab] | 38.35 ± 3.06[cd] |
| Azar 2 - | 68.45 ± 6.48 | 59.37 ± 18.84 | 63.80 ± 8.86[c] | 73.97 ± 11.91[ab] | 21.60 ± 1.62[d] | 50.27 ± 7.84 [ab] |
| Azar 2 + | 68.67 ± 2.23 | 83.54 ± 6.39 | 51.50 ± 1.17[e] | 64.92 ± 9.08[b] | 28.65 ± 0.20[c] | 43.13 ± 1.15[bc] |
| Pishgam - | 65.06 ± 9.81 | 81.26 ± 12.85 | 60.40 ± 4.59[cd] | 82.33 ± 6.08[a] | 25.54 ± 3.06[cd] | 52.33 ± 13.50 [ab] |
| Pishgam + | 63.51 ± 11.20 | 83.21 ± 8.69 | 48.70 ± 3.44[e] | 73.51 ± 11.04[ab] | 39.10 ± 2.38[b] | 20.75 ± 5.60[e] |
| Sirvan - | 69.35 ± 2.89 | 87.07 ± 4.08 | 59.56 ± 0.43[cd] | 35.75 ± 13.25[d] | 31.71 ± 4.14[c] | 26.18 ± 3.03[e] |
| Sirvan + | 78.30 ± 2.33 | 79.69 ± 7.06 | 70.47 ± 1.87[b] | 58.37 ± 9.31[bc] | 47.46 ± 4.26[a] | 27.87 ± 4.18[de] |
| Corn | 75.33 ± 5.68 | 75.30 ± 8.38 | 53.92 ± 2.43[de] | 41.94 ± 4.36[d] | 47.28 ± 7.78[a] | 46.09 ± 3.70[bc] |
| p-value | 0.177 | 0.120 | 0.0001 | 0.0001 | 0.0001 | 0.0001 |
| SEM | 3.59 | 7.53 | 2.18 | 5.28 | 2.33 | 3.58 |

WC, wheat cultivar; E, enzyme

[a-c] Means in a column, within a group, but with different superscripts are significantly (p < 0.05) different.

Values represent means ± standard deviation of the mean.

SEM, standard error of the mean.

*Bifidobacteria* and *Enterococcus* populations (p < 0.05). For *Bifidobacteria*, the Sirvan cultivar diet displayed the highest count, significantly different from diets containing Azar 2 and Pishgam cultivars (which had the lowest counts). Interestingly, only the corn diet and the Sirvan cultivar with enzyme supplementation showed a significant difference (p < 0.05), with the corn diet having a lower *Bifidobacteria* count. Similarly, wheat cultivars significantly impacted *Enterococcus* populations (p < 0.05). Diets containing Sardari and Sirvan cultivars had the highest counts, while the Azar 2 cultivar diet exhibited the lowest.

## Discussion

### Growth performance

Our study found that broilers offered diets containing various wheat cultivars displayed a decrease in FI compared to those fed a corn-based diet. Notably, supplementing these wheat

**Table 6. The effect of wheat cultivars with and without enzyme and control diet on ileum microbiota profile (log CFU/g digesta) in broilers.**

| | Ileum microbiota population | | | |
|---|---|---|---|---|
| | *Lactobacillus* | *Bifidobacteria* | *Enterococcus* | *Escherichia coli* |
| **WC** | | | | |
| Sardari | 376.00 ± 151.11 | 11.30 ± 4.21[ab] | 41.96 ± 17.30[ab] | 6.26 ± 2.07 |
| Azar 2 | 379.16 ± 135.16 | 7.81 ± 2.44[b] | 24.38 ± 11.06[b] | 4.58 ± 1.31 |
| Pishgam | 314.16 ± 111.59 | 8.66 ± 2.63[b] | 31.48 ± 13.56[ab] | 5.08 ± 1.08 |
| Sirvan | 414.16 ± 100.12 | 14.05 ± 3.67[a] | 48.08 ± 15.78[a] | 6.61 ± 1.61 |
| p-value | 0.272 | 0.018 | 0.046 | 0.097 |
| SEM | 50.78 | 1.33 | 5.81 | 0.609 |
| **E** | 399.41 ± 124.18 | 10.05 ± 3.69 | 34.13 ± 15.37 | 5.35 ± 1.50 |
| Zero | 352.33 ± 139.29 | 10.85 ± 4.36 | 38.82 ± 17.98 | 5.91 ± 1.88 |
| 0.2, g/Kg | | | | |
| p-value | 0.802 | 0.556 | 0.431 | 0.375 |
| SEM | 35.91 | 0.942 | 4.10 | 0.430 |
| **WC × E** | | | | |
| p-value | 0.254 | 0.263 | 0.256 | 0.199 |
| CV | 35.96 | 31.21 | 30.01 | 26.47 |
| Treatments | | | | |
| **WC        E** | | | | |
| Sardari— | 381.33 ± 120.08 | 11.73 ± 3.69[abc] | 43.30 ± 12.73 | 6.46 ± 2.15 |
| Sardari + | 370.66 ± 206.35 | 10.86 ± 5.49[abc] | 40.63 ± 24.11 | 6.05 ± 2.43 |
| Azar 2 - | 350.33 ± 160.46 | 9.26 ± 2.10[bc] | 29.60 ± 14.28 | 5.16 ± 1.51 |
| Azar 2 + | 208.00 ± 68.78 | 6.36 ± 2.06[c] | 19.16 ± 4.50 | 4.00 ± 1.01 |
| Pishgam - | 236.66 ± 92.51 | 6.76 ± 2.32[c] | 26.63 ± 8.35 | 4.30 ± 0.81 |
| Pishgam + | 391.66 ± 67.51 | 10.56 ± 1.07[abc] | 42.33 ± 6.11 | 5.86 ± 0.66 |
| Sirvan - | 389.33 ± 117.05 | 12.46 ± 4.61[ab] | 43.00 ± 17.77 | 5.50 ± 1.08 |
| Sirvan + | 439.00 ± 97.53 | 15.63 ± 2.19[a] | 53.16 ± 15.14 | 7.73 ± 1.26 |
| Corn | 294.66 ± 91.30 | 8.66 ± 1.85[bc] | 26.60 ± 7.87 | 5.44 ± 1.68 |
| p-value | 0.341 | 0.048 | 0.076 | 0.181 |
| SEM | 69.96 | 1.81 | 7.89 | 0.874 |

WC, wheat cultivar; E, enzyme

[a-c] Means in a column, within a group, but with different superscripts are significantly (p < 0.05) different.

Values represent means ± standard deviation of the mean.

SEM, standard error of the mean.

diets with enzymes did not significantly alter FI compared to the unsupplemented diets. This suggests that enzymes may not fully compensate for the differences in FI caused by varying wheat cultivars. These findings align with previous research where researchers also observed variations in FI based on the wheat cultivar, with enzymes failing to completely eliminate these differences [12, 21]. Similarly, a study showed reduced FI in wheat diets compared to corn and a limited effect of adding multi-carbohydrases on FI [22]. The culprit for this reduced FI is likely the presence of NSPs, particularly arabinoxylan polymers, in wheat. These NSPs are thought to have negative effects on a broiler's appetite [22]. Additionally, high digesta viscosity caused by NSPs can slow down gut passage, leading to reduced FI [23]. It's important to note that contrasting results exist. Hajati [24] observed a decrease in FI with enzyme addition in both corn and wheat diets. This discrepancy could be attributed to factors like specific wheat cultivars used, enzyme type and dosage, and broiler age. Furthermore, Cardoso et al. [23]

highlight the variation in endogenous xylanase activity among different wheat cultivars, suggesting that enzyme supplementation might be more effective in diets with high NSP content and low endogenous enzyme activity.

Interestingly, our study did not observe significant changes in DWG or FCR due to wheat cultivar or enzyme supplementation. This finding contrasts with some previous research. For example, a study reported improved FCR with enzyme addition in wheat-based diets [21], while another research observed similar benefits with xylanase supplementation in diets containing different wheat varieties [25]. However, other studies align more closely with our results. Del Alamo et al. [12] found variations in DWG and FCR among wheat cultivars, but enzymes did not eliminate these differences. This finding highlights the potential for wheat cultivar and enzyme type to interact, influencing DWG and FCR. As some studies point out, inconsistencies in these parameters can arise from variations in wheat cultivar, enzyme characteristics, and broiler factors like age and breed [1, 12]. Further research is needed to explore these interactions and identify optimal combinations for maximizing DWG and FCR in wheat-based broiler diets.

### Nutrient digestibility

Our findings revealed a significant interaction between wheat cultivar and enzyme supplementation on broiler ileal CP digestibility. The highest digestibility was observed in the unenzymed Sardari cultivar, while the lowest occurred in the Pishgam cultivar with enzyme. This aligns with the established concept that high dietary NSP content increases digesta viscosity, reducing nutrient digestibility [26, 27]. Table 3 indicates that the Sardari cultivar had the lowest soluble fiber content, potentially explaining its improved CP digestibility compared to others. Similar trends were observed by some other researchers who reported that CP digestibility was influenced by wheat cultivar and correlated with NSP content [25, 28]. However, Del Alamo et al. [12] found no significant differences in CP digestibility among wheat cultivars. These discrepancies likely stem from variations in wheat characteristics, particularly NSP levels (soluble and insoluble fiber), alongside measurement methods and broiler age. Interestingly, our study showed lower CP digestibility in corn-fed chickens compared to some wheat cultivars. This aligns with a research that observed lower ileal CP digestibility in corn compared to wheat diets [29]. The effect of enzyme supplementation also displayed interesting interactions. While enzyme addition generally reduced CP digestibility across cultivars, the Sirvan cultivar (highest NSP content) showed an increase with enzyme supplementation. This suggests that enzymes may be more effective in diets with higher NSP content due to increased available substrate for hydrolysis [23]. Previous research on enzyme effects on CP digestibility in wheat-based diets is mixed. Some studies reported improved protein digestibility [8, 30, 31] while others observed no significant effect [12, 25, 32]. These inconsistencies highlight the complex interplay between factors like intrinsic wheat properties, dietary composition, and broiler characteristics that influence the response to xylanase supplementation [1, 33].

Our study revealed a significant interaction between wheat cultivar and enzyme supplementation on broiler ileal fat digestibility. Different cultivars with and without enzymes displayed varied responses. Additionally, fat digestibility in the corn diet was lower than in most wheat diets. This aligns with previous research reporting higher ileal fat digestibility in broilers fed wheat compared to corn [29]. Similarly, another study observed a potential influence of varying NSP concentrations in wheat cultivars on fat digestibility [30]. Interestingly, adding enzymes significantly boosted fat digestibility in the Sardari and Sirvan cultivars but had no significant effect on the Pishgam and Azar 2 cultivars. This differential response is supported by studies that highlight the positive impact of enzymes on fat digestibility in wheat diets [29,

30, 34]. The underlying reason for these variations likely relates to the chemical composition of wheat, particularly NSPs, and their impact on digesta viscosity [1, 35]. This, in turn, affects the response to enzymes.

A significant interaction between wheat cultivar and enzyme supplementation was observed for the ileal digestibility of calcium and phosphorus. For phosphorus, diets containing the unenzymed Sardari cultivar, Sirvan with enzyme, and corn had the highest digestibility. Differences in wheat chemistry likely contribute to the variation observed among cultivars. Enzyme addition significantly improved phosphorus digestibility in Azar 2, Pishgam, and Sirvan cultivars but had no significant effect on the Sardari cultivar. This suggests that enzymes might be more beneficial in diets with higher NSP content due to increased available substrate for hydrolysis [23]. The positive impact of enzymes on nutrient digestibility is attributed to reduced digesta viscosity, nutrient liberation, increased cell wall permeability, and reduced endogenous amino acid secretion [26, 27, 36]. Studies support this notion, with Nortey et al. [37] reporting improved phosphorus digestibility with xylanase supplementation in a wheat millrun diet, while Liu and Kim [38] observed no effect. Mature seeds store phosphorus as phytate, primarily located in the outer layer rich in arabinoxylan, the main substrate for xylanase enzymes [39, 40]. Therefore, enzyme addition to diets high in arabinoxylan might indirectly improve phosphorus digestibility. The unenzymed Sardari, Azar 2, and Pishgam cultivars displayed the highest calcium digestibility (Table 5), likely due to their lower NSP content compared to the Sirvan cultivar (highest NSP content; Table 3). Interestingly, enzyme addition reduced calcium digestibility in the Sardari and Pishgam cultivars but had no significant effect on Azar 2 and Sirvan cultivars. This aligns with findings from Liu and Kim [38] and Nortey et al. [41] who reported no significant changes in calcium digestibility with enzyme supplementation in broiler diets.

## Intestinal microflora

The chicken gastrointestinal (GI) tract is home to a diverse community of microorganisms. The interactions between the host and the GI microbiome are crucial for the chicken's health [42]. Specifically, the gut microbiota aids in breaking down complex food molecules into simpler forms that can be absorbed by the host. It also produces essential vitamins and short-chain fatty acids, which provide energy and support gut health. Additionally, the microbiota helps to maintain a healthy immune system by stimulating the development of immune cells and creating a protective barrier against harmful microorganisms [42]. Recent research suggests a link between dietary NSPs and gut microbial composition [43]. In this study, the population of *Bifidobacteria* and *Enterococcus* in birds fed diets containing Sirvan cultivar with higher NSP content was higher than other wheat cultivars. This aligns with the established role of soluble NSPs as fermentable substrates for beneficial bacteria [44–46]. Their fermentation produces short-chain fatty acids (SCFAs) that act as an energy source for gut cells and inhibit harmful bacteria [44]. While our study didn't show a significant effect on *Lactobacillus* and *E. coli* populations, other studies reported increases in these beneficial bacteria with wheat or barley-based diets (higher NSP) compared to corn (lower NSP) [47, 48]. This suggests a potential benefit of NSPs for promoting some beneficial gut bacteria. The use of enzymes to break down NSPs yielded mixed results. Our study, along with those by Luo et al. [21] and Gao et al. [49], found no significant impact on ileal bacterial populations with enzyme supplementation. However, Liu and Kim [38] reported increased *Lactobacillus* and decreased *E. coli* with enzyme use. These inconsistencies highlight the potential influence of factors like enzyme type and NSP composition on their effectiveness [50].

## Conclusions

In summary, wheat diets generally reduced FI compared to corn and enzymes did not significantly improve it. There were no major effects on DWG or FCR. Wheat cultivar and enzyme interacted to influence nutrient digestibility. Enzymes were more effective in improving the digestibility of protein and fat in cultivars with higher NSP content. Also in this study, an increase in the population of ileum microbiota was observed as a result of using wheat cultivars with higher amounts of NSPs in the broilers' diets. Overall, the complex interplay between wheat characteristics, enzyme type, and broiler factors necessitates further research to optimize wheat-based broiler diets.

## Supporting information

**S1 File.**
(XLSX)

## Acknowledgments

The authors appreciate the head of Animal Science Department for supporting the current study.

## Author Contributions

**Conceptualization:** Seyedkamyar Seyedoshohadaei, Mehran Torki.

**Data curation:** Seyedkamyar Seyedoshohadaei, Mehran Torki, Akbar Yaghoubfar, Alireza Abdolmohammadi.

**Formal analysis:** Alireza Abdolmohammadi.

**Investigation:** Seyedkamyar Seyedoshohadaei, Mehran Torki, Akbar Yaghoubfar, Alireza Abdolmohammadi.

**Methodology:** Seyedkamyar Seyedoshohadaei, Mehran Torki, Alireza Abdolmohammadi.

**Project administration:** Mehran Torki.

**Resources:** Mehran Torki.

**Software:** Seyedkamyar Seyedoshohadaei, Alireza Abdolmohammadi.

**Supervision:** Mehran Torki.

**Validation:** Mehran Torki, Alireza Abdolmohammadi.

**Writing – review & editing:** Mehran Torki, Akbar Yaghoubfar, Alireza Abdolmohammadi.

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
