## [Decision Letter · Decision Letter 0]

28 Aug 2024

PONE-D-24-31831Interaction of wheat cultivar and enzyme on broiler growth, nutrient utilization, and gut microfloraPLOS ONE

Dear Dr. Torki,

Thank you for submitting your manuscript to PLOS ONE. After careful consideration, we feel that it has merit but does not fully meet PLOS ONE’s publication criteria as it currently stands. Therefore, we invite you to submit a revised version of the manuscript that addresses the points raised during the review process.

We look forward to receiving your revised manuscript.

Kind regards,

Ewa Tomaszewska, DVM Ph.D

Academic Editor

PLOS ONE

Journal Requirements:

Reviewers' comments:

Reviewer's Responses to Questions

**Comments to the Author**

1. Is the manuscript technically sound, and do the data support the conclusions?

Reviewer #1: Yes

2. Has the statistical analysis been performed appropriately and rigorously? 

Reviewer #1: Yes

3. Have the authors made all data underlying the findings in their manuscript fully available?

Reviewer #1: Yes

4. Is the manuscript presented in an intelligible fashion and written in standard English?

Reviewer #1: Yes

5. Review Comments to the Author

Reviewer #1: Dear authors,

Your manuscript, "Interaction of Wheat Cultivar and Enzyme on Broiler Growth, Nutrient Utilization, and Gut Microflora" examines the impact of four wheat cultivars (Sardari, Azar2, Sirvan, and Pishgam) and enzyme supplementation on the growth performance, nutrient digestibility, and ileal microbiota composition of broiler chickens. The manuscript is generally well-written and organized. However, there are a few questions and suggestions for improvement. Please address the following.

1. What are the effects of different wheat cultivars and enzyme supplementation on broiler growth performance?

2. How does enzyme supplementation impact the nutrient digestibility in broilers fed with different wheat cultivars?

3. What changes occur in the ileal microbiota composition of broilers when different wheat cultivars and enzyme treatments are used?

4. What are the limitations of your study, such as the potential impact of bird age, breed, and enzyme type on the results?

5. What are the broader implications of your work for future research and practical applications in the poultry industry?

6. What is the potential economic impact of your findings, as a key consideration for the poultry industry?

7. Please proofread the manuscript for any typographical or grammatical errors like “CO2” in line number 117, to enhance readability.

8. To enhance the manuscript’s accessibility for a broader audience, it would be beneficial to briefly explain the function of gut microbiota in birds/broilers.

6. PLOS authors have the option to publish the peer review history of their article (what does this mean?). If published, this will include your full peer review and any attached files.

Reviewer #1: **Yes: **Dr. Niaz Ali

---

## [Author Response · Author response to Decision Letter 0]

7 Oct 2024

Thank you for your valuable feedback on our manuscript. We appreciate your time and effort in reviewing our work. Your comments have been very helpful in strengthening our manuscript. We have carefully considered your suggestions and have made the necessary revisions. Please find below our responses to your specific questions.

1. What are the effects of different wheat cultivars and enzyme supplementation on broiler growth performance?

Wheat, a common ingredient in broiler diets, can vary in quality based on cultivar and growing conditions. The non-starch polysaccharide (NSP) content of wheat affects its digestibility and energy value for broilers. High NSP levels can reduce nutrient absorption and increase gut viscosity. Enzyme supplementation, such as xylanase and β-glucanase, can improve wheat digestibility by breaking down NSPs. However, in our study, DWG and FCR were not altered under the influence of wheat cultivars and supplemental enzymes. While studies have generally shown that enzyme supplementation can enhance broiler performance, the specific effects can vary depending on factors like wheat cultivar, enzyme type, and broiler age and breed. The activity of endogenous xylanase enzymes in broilers can also influence the effectiveness of supplemental enzymes (Amerah, 2015; Del Alamo et al., 2008; Cardoso et al., 2018).

2. How does enzyme supplementation impact the nutrient digestibility in broilers fed with different wheat cultivars?

Enzyme supplementation can significantly boost nutrient digestibility in broiler chickens consuming different wheat varieties, especially those with high levels of NSPs. NSPs can hinder nutrient absorption and increase gut viscosity. Enzymes like xylanase and β-glucanase break down NSPs, improving nutrient digestibility. The effectiveness of enzyme supplementation depends on factors such as the wheat cultivar, enzyme type and dosage, and the age and nutritional needs of the broiler chickens. Our findings highlight the interactive effects of wheat cultivar and enzyme supplementation on nutrient digestibility:

Protein Digestibility: Enzyme addition generally reduced CP digestibility across cultivars, however, Sirvan cultivar (highest NSP content) showed an increase with enzyme supplementation. This suggests that enzymes may be more effective in diets with higher NSP content due to increased available substrate for hydrolysis (Cardoso et al., 2018).

Fat Digestibility: Adding enzymes significantly boosted fat digestibility in the Sardari and Sirvan cultivars but had no significant effect on the Pishgam and Azar 2 cultivars. This differential response is supported by studies that highlight the positive impact of enzymes on fat digestibility in wheat diets (Kiarie et al., 2014; Smeets et al., 2018). The underlying reason for these variations likely relates to the chemical composition of wheat, particularly NSPs, and their impact on digesta viscosity (Choct et al., 2004).

Mineral Digestibility: Enzyme supplementation influenced calcium and phosphorus digestibility, with varying effects depending on the wheat cultivar.

3. What changes occur in the ileal microbiota composition of broilers when different wheat cultivars and enzyme treatments are used?

The ileal microbiota plays a key role in broiler health. The composition of this microbiota can be influenced by various factors, including diet. Wheat cultivars and enzyme treatments can significantly impact the ileal microbiota. Different wheat cultivars have varying levels of non-starch polysaccharides (NSPs), which can promote the growth of specific microbial populations, such as Bifidobacteria and Enterococcus (Apajalahti et al., 2004). Enzymes like xylanase and β-glucanase can break down NSPs, improving nutrient digestibility and indirectly influencing the ileal microbiota. Our study found that the Sirvan cultivar, with higher NSP content, increased Bifidobacteria and Enterococcus populations. These beneficial bacteria produce short-chain fatty acids, which support gut health (Bao and Choct, 2010). While our study did not observe significant changes in Lactobacillus and E. coli, other studies have reported increases in these bacteria with wheat or barley-based diets. These findings highlight the importance of carefully selecting wheat cultivars and considering enzyme supplementation to optimize broiler health and performance (Nguyen et al., 2021).

4. What are the limitations of your study, such as the potential impact of bird age, breed, and enzyme type on the results?

We appreciate the reviewer's insightful comment regarding the potential limitations of our study. As noted, the current investigation was conducted with broiler chickens from 1 to 39 days of age, using Ross 308 birds and a specific enzyme supplement, beta-endopower®. While our study provides valuable insights, it is important to acknowledge that the effects of wheat cultivars and enzyme supplementation may vary across different bird ages, breeds, and enzyme types. For instance, the sensitivity of the digestive system to dietary changes might differ during the various developmental stages of broilers. Additionally, genetic differences in digestive physiology, growth rates, and metabolic characteristics among different breeds could influence their responses to dietary factors. Furthermore, the specific activities and specificities of different enzyme preparations may vary, potentially affecting their impact on nutrient digestibility and microbiota composition.

5. What are the broader implications of your work for future research and practical applications in the poultry industry?

In terms of practical applications, our findings can inform the formulation of poultry diets, particularly those containing wheat. Feed manufacturers can leverage this information to select wheat cultivars with optimal nutritional properties and to determine the appropriate levels of enzyme supplementation to maximize broiler growth and well-being. Moreover, our insights into the interplay between diet, microbiota, and broiler health can contribute to the development of strategies to reduce the reliance on antibiotics, promoting more sustainable and resilient poultry production systems.

6. What is the potential economic impact of your findings, as a key consideration for the poultry industry?

The poultry industry can significantly benefit financially by strategically selecting wheat cultivars and using enzyme supplements. These approaches improve nutrient absorption, decrease intestinal stickiness, and foster a healthier gut environment. This results in better feed efficiency, lower feed costs, faster growth, and reduced mortality. Consequently, poultry producers can increase their profits and stay competitive. Furthermore, the positive environmental effects of decreased nutrient waste contribute to more sustainable poultry farming practices.

7. Please proofread the manuscript for any typographical or grammatical errors like “CO2” in line number 137, to enhance readability.

Thank you for your careful review. We have thoroughly examined the manuscript for any typographical or grammatical errors, including the specific concern regarding CO2 in line 137. We have made the necessary corrections to ensure the highest quality of the text.

8. To enhance the manuscript’s accessibility for a broader audience, it would be beneficial to briefly explain the function of gut microbiota in birds/broilers.

Thank you again for your valuable comments on our manuscript. In response to your recommendation, we have added a brief explanation of the gut microbiota's function to the discussion section. This addition aims to enhance the manuscript's accessibility for a broader audience (page 14, lines 298-303).

References 

Amerah AM. Interactions between wheat characteristics and feed enzyme supplementation in broiler diets. Anim Feed Sci Tech. 2015;199:1-9. https://doi.org/10.1016/j.anifeedsci.2014.09.012

Apajalahti J, Kettunen A, Graham H. Characteristics of the gastrointestinal microbial communities, with special reference to the chicken. Worlds Poult Sci J. 2004;60(2):223-32. https://doi.org/10.1079/WPS200415

Bao YM, Choct M. Dietary NSP nutrition and intestinal immune system for broiler chickens. Worlds Poult Sci J. 2010;66(3):511-8. https://doi.org/10.1017/S0043933910000577

Cardoso V, Fernandes EA, Santos HM, Maçãs B, Lordelo MM, Telo da Gama L, Ferreira LM, Fontes CM, Ribeiro T. Variation in levels of non-starch polysaccharides and endogenous endo-1, 4-β-xylanases affects the nutritive value of wheat for poultry. Br Poult Sci. 2018;59(2):218-26. https://doi.org/10.1080/00071668.2018.1423674

Choct M, Kocher A, Waters DL, Pettersson D, Ross G. A comparison of three xylanases on the nutritive value of two wheats for broiler chickens. Br J Nutr. 2004;92(1):53-61. https://doi.org/10.1079/BJN20041166

Del Alamo AG, Verstegen MW, Den Hartog LA, De Ayala PP, Villamide MJ. Effect of wheat cultivar and enzyme addition to broiler chicken diets on nutrient digestibility, performance, and apparent metabolizable energy content. Poult Sci. 2008;87(4):759-67. https://doi.org/10.3382/ps.2007-00437

Kiarie E, Romero LF, Ravindran V. Growth performance, nutrient utilization, and digesta characteristics in broiler chickens fed corn or wheat diets without or with supplemental xylanase. Poult Sci. 2014;93(5):1186-96.

Nguyen HT, Bedford MR, Wu SB, Morgan NK. Soluble non-starch polysaccharide modulates broiler gastrointestinal tract environment. Poult Sci. 2021:101183. https://doi.org/10.1016/j.psj.2021.101183

Smeets N, Nuyens F, Van Campenhout L, Delezie E, Niewold TA. Interactions between the concentration of non-starch polysaccharides in wheat and the addition of an enzyme mixture in a broiler digestibility and performance trial. Poult Sci. 2018;97(6):2064-70. https://doi.org/10.3382/ps/pey038

---

## [Decision Letter · Decision Letter 1]

14 Oct 2024

Interaction of wheat cultivar and enzyme on broiler growth, nutrient utilization, and gut microflora

PONE-D-24-31831R1

Dear Dr. Mehran Torki,

We’re pleased to inform you that your manuscript has been judged scientifically suitable for publication and will be formally accepted for publication once it meets all outstanding technical requirements.

Kind regards,

Ewa Tomaszewska, DVM Ph.D

Academic Editor

PLOS ONE

Additional Editor Comments (optional):

Reviewers' comments:

Reviewer's Responses to Questions

**Comments to the Author**

1. If the authors have adequately addressed your comments raised in a previous round of review and you feel that this manuscript is now acceptable for publication, you may indicate that here to bypass the “Comments to the Author” section, enter your conflict of interest statement in the “Confidential to Editor” section, and submit your "Accept" recommendation.

Reviewer #1: All comments have been addressed

2. Is the manuscript technically sound, and do the data support the conclusions?

Reviewer #1: Yes

3. Has the statistical analysis been performed appropriately and rigorously? 

Reviewer #1: Yes

4. Have the authors made all data underlying the findings in their manuscript fully available?

Reviewer #1: Yes

5. Is the manuscript presented in an intelligible fashion and written in standard English?

Reviewer #1: Yes

6. Review Comments to the Author

Reviewer #1: Dear Authors,

Thank you for addressing my suggestions. The manuscript has significantly improved and now appears to be suitable for publication.

7. PLOS authors have the option to publish the peer review history of their article (what does this mean?). If published, this will include your full peer review and any attached files.

Reviewer #1: **Yes: **Dr. Niaz Ali

---

## [Editor Report · Acceptance letter]

29 Oct 2024

PONE-D-24-31831R1 

PLOS ONE

Dear Dr. Torki, 

I'm pleased to inform you that your manuscript has been deemed suitable for publication in PLOS ONE. Congratulations! Your manuscript is now being handed over to our production team.

Kind regards, 

on behalf of

Professor Ewa Tomaszewska 

Academic Editor

PLOS ONE